# Review of Application and Innovation of Geotextiles in Geotechnical Engineering

**DOI:** 10.3390/ma13071774

**Published:** 2020-04-10

**Authors:** Hao Wu, Chongkai Yao, Chenghan Li, Miao Miao, Yujian Zhong, Yuquan Lu, Tong Liu

**Affiliations:** 1School of Highway, Chang’an University, Xi’an 710064, China; WCK@CHD.EDU.CN (H.W.); yaochongkai@chd.edu.cn (C.Y.); lichenghan@chd.edu.cn (C.L.); miaomiao@chd.edu.cn (M.M.); zhongyujian@chd.edu.cn (Y.Z.); luyuquan@chd.edu.cn (Y.L.); 2School of science, Xi’an University of Architecture and Technology, Xi’an 710055, China

**Keywords:** geotextiles, geotechnical engineering, natural geotextiles, application, innovation

## Abstract

Most geotextiles consist of polymers of polyolefin, polyester or polyamide family, which involve environmental problems related to soil pollution. Geotextiles can be used for at least one of the following functions: Separation, reinforcement, filtration, drainage, stabilization, barrier, and erosion protection. Due to the characteristics of high strength, low cost, and easy to use, geotextiles are widely used in geotechnical engineering such as soft foundation reinforcement, slope protection, and drainage system. This paper reviews composition and function of geotextiles in geotechnical engineering. In addition, based on literatures including the most recent data, the discussion turns to recent development of geotextiles, with emphasis on green geotextiles, intelligent geotextiles, and high-performance geotextiles. The present situation of these new geotextiles and their application in geotechnical engineering are reviewed.

## 1. Introduction

Geosynthetics are products made of synthetic or natural polymeric materials, which are used in contact with soil or rock and/or other geotechnical materials. Geosynthetics mainly include geotextile, geogrid, geocell, geonet, geomembrane, erosion control mat, geosynthetic clay liner, and geo-composite [1]. Geotextiles are the most widely used geosynthetics [2]. The first reported employment of geotextiles can be considered to be the nylon bags filled with sand used in the Dutch Delta Works in 1956 [3]. In the last 60 years, geotextiles were widely used in geotechnical engineering. Geotextiles can be used for at least one of the following functions in geotechnical engineering: Separation, filtration, drainage, reinforcement, stabilization, barrier, and erosion protection [4].

At present, more than 1.4 billion square meters of geotextiles are used every year, and the trend is on the rise. About 98% geotextiles consist of non-degradable polymers from the polyolefin, polyester, or polyamide family. The long-term use of geotextiles, because of quite a few environmental factors, such as wind, moisture, friction, and ultraviolet radiation, may cause the disintegration of synthetic polymer, resulting in the accumulation of micro plastics in the surrounding environment [5]. In addition, the application of geotextiles in geotechnical engineering may encounter complex environmental conditions [6], such as complex acid-base conditions, and then the performance requirements of geotextiles will be higher [7].

Therefore, geotextiles should develop towards high performance and multi-function. With the popularity of green concept, a multitude of scholars have studied the possibility of natural geotextiles replacing nonbiodegradable polymer geotextiles [5,8]. At present, natural geotextiles can replace synthetic geotextiles in 50% of all applications [9,10]. With the integration of optical fiber sensor into geotextiles, geotextiles are also developing towards the field of multi-functional intelligence [11,12]. When the intelligent geotextiles are applied to the reinforcement of the geotechnical structure, it can also carry out the health monitoring of the geotechnical structure, so as to ensure that the geotechnical structure and its locations with high failure and damage risk can be found in the early stage, which is conducive to prevention and maintenance in advance [13,14]. With the progress of fiber materials, a number of new high-performance geotextiles have emerged, including wicking geotextiles [15,16] and basalt fiber needling geotextiles [17].

Geotextiles are considered as cost effective, durable, and easy to use. Nowadays, geotextiles are of ever-growing indispensability in geotechnical engineering. This paper reviews composition and function of geotextiles in geotechnical engineering. In addition, based on literatures including the most recent data, the discussion turns to recent development of geotextiles, with emphasis on green geotextiles, intelligent geotextiles, and high-performance geotextiles. The development of these new geotextiles and their application in geotechnical engineering are reviewed.

## 2. Composition and Function of Geotextiles

### 2.1. Base Material of Synthetic Geotextiles

The selection of geotextile materials must be adequate to the actual situation of the project, not only considering the performance requirements of geotextiles in the site, but also considering the product cost. At present, the basic material of geotextiles is mainly synthetic fiber. Polypropylene (PP) is the most frequently used material in connection with geotextiles, followed by polyethylene terephthalate (PET) and polyethylene (PE) [5]. Figure 1 illustrates the most common polymers used as geotextiles and SEM image.

Polypropylene is one of the most widely used fibers for geotextiles because of its low cost, acceptable tensile properties, and chemical inertness. This fiber has an additional advantage because of its low density, which results in an extremely low cost per volume. The main disadvantage of polypropylene is its poor sensitivity to UV. In addition, at high temperature, its performance is easy to deteriorate, demonstrating poor creep characteristics [18]. Table 1 summarizes the properties of the most common polymers used as geotextiles.

Another major synthetic fiber for geotextiles is PET. It has excellent tensile properties and high creep resistance. The geotextiles made of polyester fiber can be used at high temperature. The main disadvantage of polyester fiber is that it is easy to hydrolyze and degrade in the soil with a pH value more than 10 [19].

Although polyethylene fiber is usually used to make geomembranes, due to the lack of supply of polyethylene fiber, it has become a rarely used solution in the field of geotextiles. Polyamide is also rarely used in geotextiles due to its low cost and poor comprehensive performance [20].

Additives are usually added to enhance the performance of geotextiles, such as antioxidants, hindered amine light stabilizer, UV absorbers and stabilizers, Long term thermal stabilizer, processing modifiers, flame retardants, lubricants, and antibacterial agents [23].

### 2.2. Function of Geotextiles

#### 2.2.1. Geotextiles Used in Separation

The separation function of geotextiles refers to a geotextile can separate two kinds of materials with different properties, avoid mixing with each other, and lose the integrity and structural integrity of various materials. In Figure 2, when stone aggregates are placed on fine-grained soil, both mechanisms will occur at the same time over time. One is that the fine soil in the lower layer tries to enter the void of the aggregate, thus destroying its drainage capacity; the other is that the aggregate in the upper layer tries to invade the fine soil, thus destroying the strength of the aggregate [24]. This usually happens without the use of geotextiles.

Figure 3 demonstrates the comparison of pavement with or without geotextiles. The separation function of geotextiles can effectively prevent the pumping effect created by dynamics [25,26]. Some of the applications areas are [27]:Between subgrade and stone base in unpaved and paved roads and airfieldsBetween subgrade in railroadsBetween landfills and stone base coursesBetween geomembranes and sand drainage layersBeneath sidewalks slabsBeneath curb areasBeneath parking lotsBeneath sport and athletic fields

#### 2.2.2. Geotextiles Used in Filtration

In an internally unstable soil, seepage may cause the phenomenon of suffusion; the transport of fine particles by seepage flow is accompanied by a collapse of the soil structure [28,29,30]. Because geotextiles have positive permeability and air permeability, they can be placed in the soil structure to allow the liquid in the soil to pass through and discharged, and play a role of soil conservation, which can effectively prevent the loss of soil particles, fine sand, and small stones in the upstream, prevent soil damage, and effectively avoid the phenomenon of suffusion [31,32,33]. The mechanism is demonstrated in Figure 4. The filtration function of geotextiles is also widely used in geotechnical engineering. As an example, geotextiles are used to prevent soil particles from migrating and infiltrating drainage aggregates or drainage pipes, while maintaining the normal operation of drainage systems; laying geotextiles under a riprap protective layer and other protective materials on coasts and riverbanks can prevent soil erosion and river bank collapse.

#### 2.2.3. Geotextiles Used in Drainage

Geotextiles have favorable water conductivity, which are used as drainage channel. The water in the soil structure in the geotextiles can be collected and slowly discharged along the geotextiles (Figure 5). At present, geotextiles have been widely used in underground drainage, subgrade drainage, retaining wall drainage, and other drainage works (Figure 6) [26,34].

#### 2.2.4. Geotextiles Used in Reinforcement

Geotextiles are placed in the interior of the soil as reinforcing materials, which combine with the soil to form a reinforced composite soil. Compared with the unreinforced soil, the strength and deformation performance of the reinforced composite soils are improved obviously (Figure 7). There are three crucial mechanical properties of geotextiles used for reinforcement: Tensile modulus, tensile strength, and surface friction. The reinforcement function of geotextiles is the most commonly used in geotechnical engineering. As shown in Figure 8, it has been widely used in reinforcing paved and unpaved roads and railroads, reinforcing walls, berms, and slopes and reinforcing soft soil foundations [6,36].

In addition to the above several main applications of geotextiles, the functions of geotextiles include barrier and erosion protection. The anti-seepage function of geotextiles is usually used in reservoir anti-seepage, tunnel anti-seepage, and landfill anti-seepage [33,37]. The protective effect of geotextiles is generally through laying geotextiles, which can reduce soil loss caused by rainfall impact and surface water runoff [9,38]. Generally speaking, geotextiles play more than two functions at the same time, when used in geotechnical engineering.

### 2.3. Analysis of the Application of Geotextiles

Geotextiles have played a crucial role in the field of geotechnical engineering and demonstrated a steady growth in global market demand. The global geotextile market size is estimated to be 4.6 billion US dollars in 2019, and the compound annual growth rate is expected to be 11.9%. Driven by China, India, and other countries, the Asia Pacific region is the largest geotextile market. Due to the high demand for geotextiles in the country’s modern infrastructure development projects, China accounted for 45.7% of the Asia Pacific geotextile market in 2019. As shown in the Figure 9, the major applications of geotextiles include road construction, erosion prevention, and drainage systems.

With the increasing demand of geotextiles, the geotextiles have also been innovated. With the development of green concept, the natural geotextiles made of natural fibers are worth considering. Intelligent geotextiles are also the trend of geotextile development. At present, the integration of optical fiber sensors into geotextiles can make geotextiles have the functions of reinforcement, structural safety monitoring, and early warning [12,39,40]. The latest development of natural geotextile, intelligent geotextile, and high-performance geotextile will be summarized in detail below.

## 3. Green Geotextiles

Most geotextiles are made of PP, PET, and PE, which are non-degradable polymers. In the long-term use process, because of external environmental factors, such as wind, water, friction, and UV radiation, it may cause the collapse of synthetic polymers, resulting in the accumulation of micro plastics in the surrounding environment [44,45,46]. Secondly, the disintegration of the polymer will cause the additives to filter from the geotextile, which will pollute the environment [23]. The properties of geotextiles made of natural fibers meet the needs of most applications, and have low cost and biodegradability. Therefore, natural geotextiles have the potential to replace synthetic geotextiles in quite a few applications of geotextiles [9,47].

### 3.1. Base Material of Natural Geotextiles

There are three kinds of natural fibers: Plant fibers, animal fibers, and mineral fibers. Among these fibers, plant fibers have become the first choice for natural geotextiles because of rich sources, easy extraction, low cost, and superior performance [48,49].

The properties of natural fibers are determined by the composition and location of the components [50]. The main components of natural fibers are cellulose, hemicellulose, lignin, and pectin [51]. The content of each component in each natural fiber is different. Cellulose, hemicellulose, and lignin are the basic components that determine the physical properties of fibers [52]. Cellulose is the hardest and strongest organic component in fiber [53]. Hemicellulose has an open structure, which contains an army of hydroxyl and acetyl groups. Therefore, hemicellulose is partially soluble in water and has hygroscopicity. Lignin is mainly aromatic, phenylpropane unit polymer. The service life of natural geotextile depends on the content of cellulose and lignin in the component fiber; the higher the content, the higher the durability [54]. Figure 10 summarizes chemical structure and properties of main components in plant fiber.

The properties of natural fibers vary with different types. Natural geotextiles usually choose natural fibers with high mechanical properties as raw materials. Table 2 outlines the composition and properties of natural fibers commonly used to make natural geotextiles. Among these fibers, jute and coir fibers have become the best research materials for natural geotextiles due to their superior performance, and several commercial products are available.

### 3.2. Performance of Natural Geotextiles in Geotechnical Engineering

In recent years, natural geotextiles have been greatly developed. A multitude of scholars have studied the application and performance of natural geotextiles in geotechnical engineering [5,8,48,56]. The author sorts out and analyzes the application of natural geotextiles in geotechnical engineering in recent years. Figure 11 demonstrates the common natural geotextiles at present.

Soil erosion refers to the process that soil is stripped and moved by wind or surface water due to inadequate or inappropriate vegetation coverage [56,66]. Soil degradation caused by erosion has a serious adverse impact on environmental sustainability and agricultural productivity. In order to prevent the soil from further degradation, geotextiles are used as the soil cover to provide temporary protection for the soil, which can effectively control erosion until the soil is stabilized by vegetation. Geotextiles made of natural fibers have high efficiency in erosion control [60]. Table 3 summarizes natural geotextiles used for protection and its protective effect. Compared with synthetic geotextiles, natural geotextiles are more effective in controlling erosion and have better soil adhesion [57]. More importantly, due to the biodegradability of natural geotextiles, the accumulation of microplastics in the environment can be avoided. Moreover, the short life and hygroscopicity of natural geotextiles have little effect on erosion control [66]. As long as vegetation is established on site, geotextiles become redundant in erosion control, so the service life of natural geotextile is about two to three years, which has little impact on erosion control. The water absorption and water storage of natural geotextiles are considered as a positive aspect [65]. Firstly, high water absorption reduces soil runoff caused by heavy rain. Secondly, the stored water can be released to the ground stably in dry weather, which creates an ideal environment for plant growth. Because synthetic geotextiles hardly absorb water, it is considered that natural geotextiles are more suitable for erosion prevention than synthetic geotextiles. In addition, the decomposition products of natural geotextiles are rich in soil nutrients, which can be used as fertilizer for new vegetation [62].

Natural geotextiles have high initial tensile strength. Used for road construction, it can be effectively reinforced at the initial stage of road use [74]. The study on the bearing capacity of jute geotextiles reinforced and unreinforced roads found that compared with unreinforced roads, the bearing capacity of jute geotextile reinforced and unreinforced roads increased by about 1.5–7 times [75]. Due to the biodegradability of natural fibers, the performance of natural geotextiles will decrease with the passage of time, which will lead to the failure of later reinforcement when used in road construction [69]. In recent years, scholars have found that the long-term durability of natural geotextiles is not particularly crucial for strengthening the stability of rural highway subgrade. In India, a growing amount of natural geotextiles are used to strengthen the subgrade of rural roads [76]. This is because the natural geotextiles have high initial tensile strength, which makes the subgrade bear low stress during the construction and operation period of the road. Then, with the passage of time, the consolidation and compaction of the subgrade soil occur under the traffic load, so the bearing capacity is enhanced. After the degradation of the natural geotextile, the bearing capacity of the subgrade meets the use requirements [77]. Table 4 summarizes natural geotextile used for reinforcement and its effect.

Natural geotextiles can also be used for medium and short term filtration or drainage, and natural fibers have the ability to remove heavy metals [78,79]. Abbar et al. [80] Studied the effect of linen geotextile on soluble heavy metals using filters filled with sand and flax geotextile. The results demonstrated that flax geotextile improved the ability of the filter to retain soluble metals and attach heavy metals (Figure 12). Therefore, natural geotextiles have obvious advantages in filtering soluble heavy metals. However, the hygroscopicity of plant fibers involves the key problems of water absorption and fiber expansion. It is found that jute geotextile, as drainage medium, is feasible and economical dealing with drainage problems encountered in geotechnical engineering [63]. Due to the high-water absorption, the pore diameter of the filter is significantly reduced. In the worst case, this will lead to filter plugging and complete loss of permeability. The hygroscopicity and service life of natural geotextiles limit the filtration or drainage function of natural geotextiles.

From the above application of natural geotextiles, it can be found that the early biodegradation of natural geotextiles is the primary problem that limits the further function of natural geotextiles [5]. Much of the research in natural geotextiles in the last two decades has examined how to improve the durability of natural geotextiles. Table 5 demonstrates the methods to improve the performance of natural geotextiles in recent years. At present, the main methods enhancing the properties of natural geotextile are to add a certain quantity of synthetic fiber, or to modify the natural geotextile chemically. Adding a certain amount of synthetic fiber can improve the physical and mechanical properties of natural geotextile. These natural fiber/polymer composite geotextiles, such as jute/PET geotextiles, jute/PP geotextiles [60,81], and nettle/ polylactic acid geotextiles [82], demonstrate promising performance in various tests. Chemical modification or use of adhesion promoters can also be interesting paths to improve the overall mechanical properties. La Mantia et al. [83] summarized the methods of chemical modification of natural fiber: Alkali treatment, acetylation, treatment with stearic acid, benzylation, TDI treatment, peroxide treatment, anhydride treatment, permanganate treatment, silane treatment, isocyanate treatment, and plasma treatment. A multitude of researchers have studied chemical modified geotextiles, such as the chemical degradation resistance of jute geotextiles treated by esterification is improved [84,85], physical properties of jute geotextiles treated with laccase are improved, and the anti-puncture ability of jute geotextiles treated with alkali is better [86]. 

## 4. Intelligent Geotextiles

Fiber optical sensor is a new type of sensor based on optical fiber developed in the mid-1970s [92,93]. It has the advantages of superior electrical insulation, strong anti-electromagnetic interference, high sensitivity, and easy to realize remote monitoring of the measured signal [94,95,96]. The optical fiber sensor is integrated into the geotextiles by using a number of warp knitting fixed positions between the sensor and the fabric. When geotextiles are applied to the stability and reinforcement of geotechnical structures such as dams, railways, embankments, and slopes, optical fiber sensors can provide other functions for geotextiles, such as monitoring the changes of mechanical deformation, temperature, humidity, and pore pressure to monitor the health of geotechnical structures, so as to ensure the early detection of high failure and failure risk of geotechnical structures and their locations, which is advantageous prevent or repair in advance [97]. Therefore, intelligent geotextiles have a superior prospect.

### 4.1. Development of Intelligent Geotextiles

Fiber optic sensor based on Fiber Bragg grating (FBG) is the first sensor applied to geotextiles [98,99,100]. The working principle and sensing principle of FBG are shown in Figure 13. However, this technology is a quasi distributed measurement technology, which only provides strain information along a finite number of points of an optical fiber, resulting in the FBG based optical fiber sensor monitoring system can only measure the quasi distributed strain of a limited length, which cannot meet the monitoring needs of hundreds of meters long embankment, dam, railway, slope, and other geotechnical structures. Secondly, the FBG sensor uses silica fiber as sensor medium, and the ultimate strain of common silica optical fiber is only about 5%. The cost of FBG sensor is relatively high, around than 20 US dollars for each point sensor [101]. Therefore, its high cost and vulnerability also limit the use of this fiber-optic sensor in intelligent geotextiles [102,103].

In order to meet the needs of large-area geotechnical structure monitoring, distributed optical fiber sensors came into being. Brillouin scattering is a kind of light scattering phenomenon caused by the interaction between the light wave incident into the fiber and the elastic sound wave in the fiber [100,105,106]. Optical fiber distributed Brillouin scattering sensor uses its backscattering signal to measure the external physical quantities. Geotextiles based on this kind of sensor (Figure 14), it can not only strengthen the geotechnical structure, but also meet the monitoring needs of large-area geotechnical structure.

When the mechanical deformation (strain >1%) of the above two kinds of intelligent geotextiles occurs, the optical fiber sensor is likely to fail, and the brittleness of silica fiber must be protected by expensive cable. Polymer optical fiber sensor integrated geotextile (Figure 15) provides a solution for distributed measurement of high strain. POF optical time domain reflectometer technology is based on the principle of increasing the level of scattered light at the place where strain is applied to POF [107,108,109]. Liehr et al. [110] found that with OTDR technology, up to 40% of the strain distribution can be measured using standard polymethyl methacrylate POF. Compared with the above-mentioned sensors, POF with high elasticity, high fracture strain, and measurement strain (higher than 40%) has more advantages as the integration of sensors on geotextile, especially for the monitoring of high mechanical deformation (such as slope with landslide risk) in a small area. Low loss perfluoropof can extend the measurement range to 500 m. POF sensor can be directly processed by textile machinery, and can be embedded in the manufacturing process of fabric structure (multi axis fabric).

In addition to the fiber optic sensor integrated geotextile, other advanced materials are also promoting the development of intelligent geotextile, such as graphene coating. Graphene can give geotextiles excellent properties, including excellent thermal conductivity (large thermal conductivity), excellent conductivity, high strength (weight reduction), stability, self repair, hydrophobicity, antibacterial performance (to prevent filter plugging), and temperature/pressure/humidity sensing function (temperature, pressure change, and leakage location detection) [111]. It can be predicted that more intelligent geotextiles will emerge in the near future and play a greater role in geotechnical engineering.

### 4.2. Application of Intelligent Geotextiles in Geotechnical Engineering-case Study

#### 4.2.1. Railway Embankment Monitoring Near Chemnitz

German STFI and Italian Alpe Adria Textile [112] have applied integrated geotextiles based on the POF OTDR technology to railway embankments near the German city of Kemnes (Figure 16). Fiber optic integrated geotextiles were intact during the rigorous installation process and were regularly questioned after the embankment reconstruction in 2007 to check the occurrence of embankment deformation or settlement that may be caused by railway transportation. The field experiment proves the feasibility of using optical fiber integrated geotextile to reinforce and monitor railway embankment at the same time. In addition, the sensor fabric can also withstand the installation of heavy machinery in the construction site without any damage. With the exception of a small and uniform increase in attenuation along the fiber, no negative long-term effects occurred within 14 months of installation.

#### 4.2.2. Field Test to Evaluate the Slope Stability at Belchatow in Poland

The scope of this field test is to assess the possibility of investigating and observing creep and landslide slopes using optical fiber integrated geotextiles. The chosen fiber is a standard PMMA POF, which is integrated in the geotextile and is able to measure distributed strain behavior on a length of 100 m. Standard PMMA POF can observe strain up to 40%. Through the evaluation of the length change of the sensing optical fiber, the position and the creep velocity of the creep landslide crack were determined. The results show that the creep velocity of the slope was constant, and the average velocity was about 2 mm per day. The test results prove the applicability of intelligent geotextiles in structural health monitoring.

#### 4.2.3. Field Tests to Evaluate the Benefit of a Geotechnical Reinforcement of a Slope Failure at Zimmersrode

Field tests were carried out at an old mining site in Zimmersrode [113], located about 50 km south of Kassel, Germany. The intelligent geotextiles that the optical fiber sensors integrated first in a thin rope-like geotextile and then placed between two layers of non-woven filter mats were used. The aim was to collect valuable information as well as the performance of the intelligent geotextiles against the weather and mechanical straining. The test results prove POF integrated geotextile provides a solution for distributed measurement of high strain, and no negative influence of moisture on sensor fibers.

## 5. High Performance Geotextiles

With the expansion of the application of geotextiles and the complexity of the environment, high-strength, multi-functional, and high-performance geotextiles have become the development direction of geotextiles.

At present, the main methods to improve the performance of geotextiles are additives, chemical modification, and composite geotextiles [114,115]. It is a shared method to use additives to make up the performance defects of geotextiles, such as stabilizers and antioxidants, but an army of additives are polluting the environment. The modification of geotextile is also a crucial means to expand the application field of geotextile and enhance the added value. The modified geotextile has higher strength and anti-degradation. As an example, grafting chitosan or cysteine covalently onto acrylic modified PP geotextile can make the geotextiles have the ability to capture heavy metals while draining or filtering [116]. Composite geotextiles are usually made of a mixture of various fibers, which combine the best functions of different materials.

Finding suitable high-performance fibers has always been a crucial research direction of high-performance geotextiles. The strength and modulus of inorganic fiber such as glass fiber [117], basalt fiber [118], and carbon fiber [119,120,121,122,123] are obviously higher than that of synthetic fiber. However, due to the high production cost, a large area of inorganic fiber on geotextiles is limited. The glass fiber composite geotextiles, made of glass fibers and short fiber needled non-woven fabric, have the characteristics of high water-permeability, excellent anti filtration, and wear resistance [124,125]. Basalt fiber geotextiles has the advantages of environmental protection and high temperature resistance [126,127]. In addition to the above several fiber materials, there are an army of fiber materials, which have the possibility of being used as high-performance geotextile raw materials, among which the most desirable one is nanofiber. Nanofibers are one of the most advanced materials, which can be easily designed into high-performance materials with unique performance, and can greatly enhance the performance of existing geotextiles [128,129,130,131].

## 6. Conclusions

In recent decades, geotextiles have been widely used in geotechnical engineering, and the global market demand is growing steadily. Geotextiles are mostly composed of polyolefin, polyester, or polyamide series polymers, and additives are usually added to enhance the performance of geotextiles. Geotextiles can be used for at least one of the following functions in geotechnical engineering: Separation, filtration, drainage, reinforcement, barrier, and erosion protection. Geotextiles play an increasingly crucial role in geotechnical engineering, but there are also quite a few potential problems: A slice of geotechnical engineering put forward higher requirements for geotextiles, the degradation of geotextiles in the application process and potential environmental pollution. These are worthy of our attention.

Based on the literature review and the latest data, the development trend of geotextiles is discussed. The natural geotextile conforms to the green concept, which is conducive to reducing environmental pollution. Natural fiber geotextile is mainly composed of plant fiber. In a slice of short and medium-term applications of geotechnical engineering, it can replace the traditional geotextile naturally. Especially in the application of erosion control, the water absorption and storage of natural geotextile are positive aspects. Durability limits the application of natural fiber geotextile. At present, the durability of natural geotextile can be effectively improved by adding quite a few synthetic fibers and chemical modification.

When the intelligent geotextile is applied to the reinforcement of the geotechnical structure, the health monitoring of the geotechnical structure can be carried out, so as to ensure that the geotechnical structure and its location with high failure and damage risk can be found in the early stage, which is conducive to prevention and maintenance in advance. The monitoring ability of the intelligent geotextile depends on the characteristics of the sensor. At present, the fiber-optic sensors integrated into geotextile mainly include fiber-optic sensors based on Fiber Bragg grating, fiber-optic sensors based on Brillouin scattering, and polymer fiber-optic sensors, which have their own advantages and disadvantages. In contrast, POF sensors with high elasticity, high fracture strain, and measuring strain (higher than 40%) are more suitable for intelligent geotextile. In addition to optical fiber sensors, there are other technologies, such as graphene technology, which can make geotextiles intelligent.

High performance geotextile is always a crucial direction of geotextile development. At present, it is mainly to add additives and modify geotextile to make up for the performance defects of geotextile. Secondly, the geotextile with excellent properties can be made from high performance fiber, such as glass fiber or basalt fiber. In the future, it is possible to design geotextiles with unique and excellent properties by applying nanofibers to geotextiles

## Figures and Tables

**Figure 1 materials-13-01774-f001:**
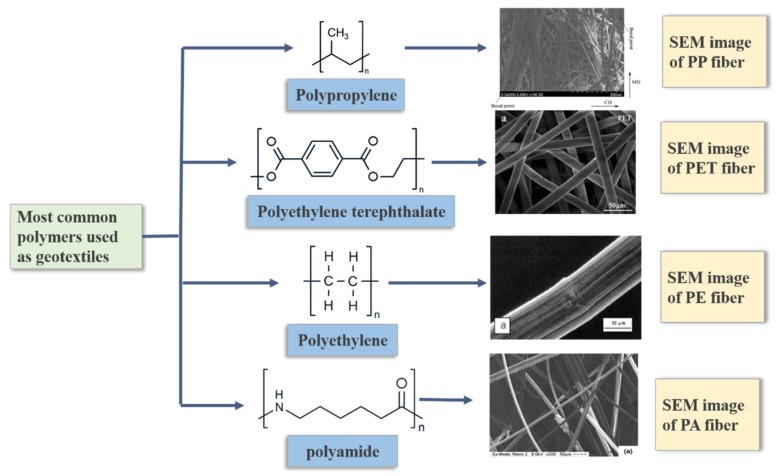
Most common polymers used as geotextiles and SEM image.

**Figure 2 materials-13-01774-f002:**
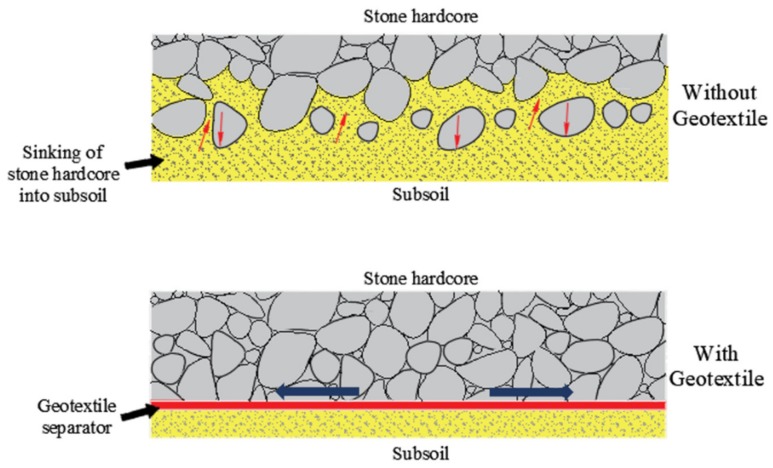
Separation function of geotextiles.

**Figure 3 materials-13-01774-f003:**
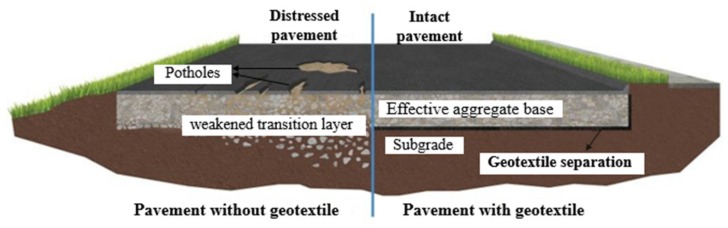
Comparison of pavement with or without geotextiles.

**Figure 4 materials-13-01774-f004:**
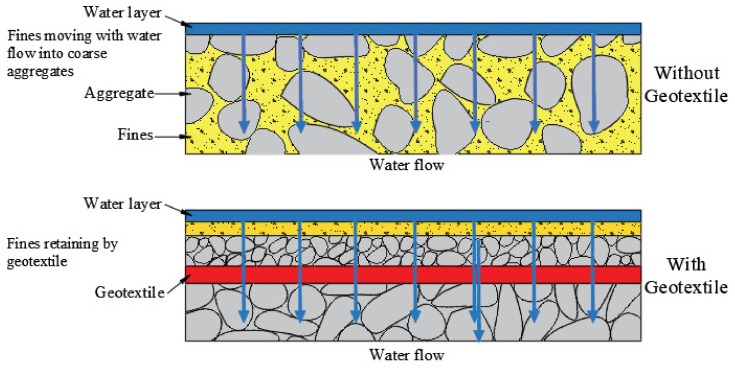
Filtration function of geotextiles.

**Figure 5 materials-13-01774-f005:**
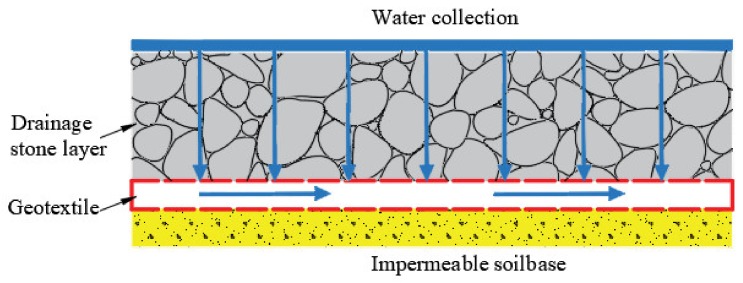
Drainage function of geotextiles.

**Figure 6 materials-13-01774-f006:**
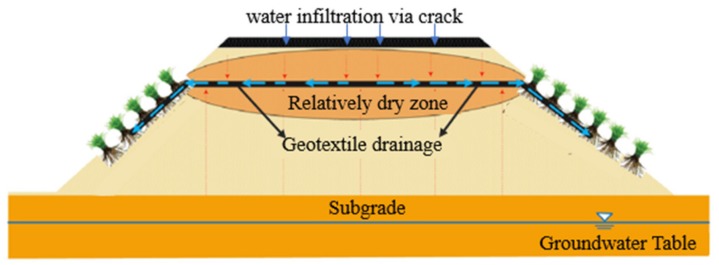
Subsurface drainage design with the wicking geotextiles [35].

**Figure 7 materials-13-01774-f007:**
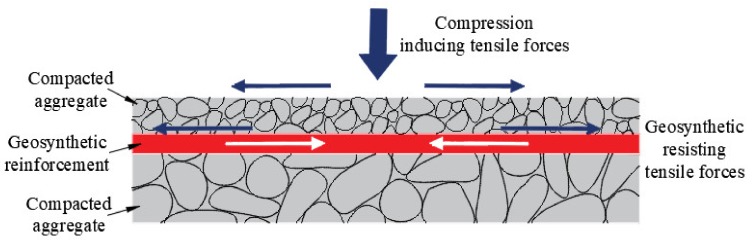
Reinforcement function of geotextiles.

**Figure 8 materials-13-01774-f008:**
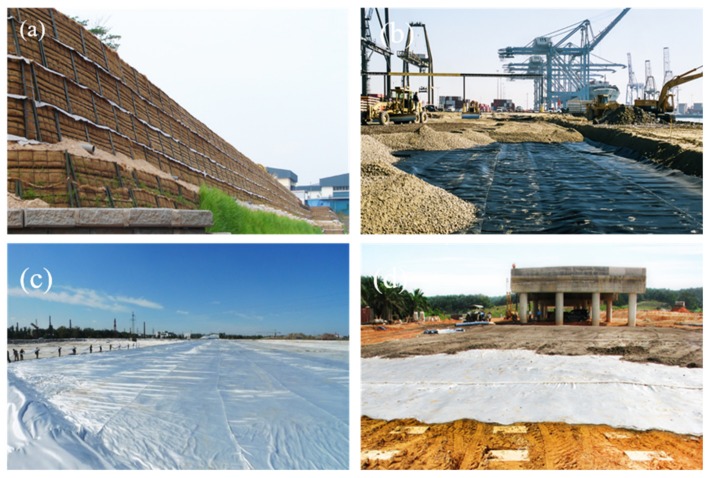
Some applications of geotextile reinforcement: (**a**) reinforcement of slopes; (**b**) reinforcement of embankment; (**c**) reinforcement of soft soil foundation; and (**d**) reinforcement of load transfer platforms.

**Figure 9 materials-13-01774-f009:**
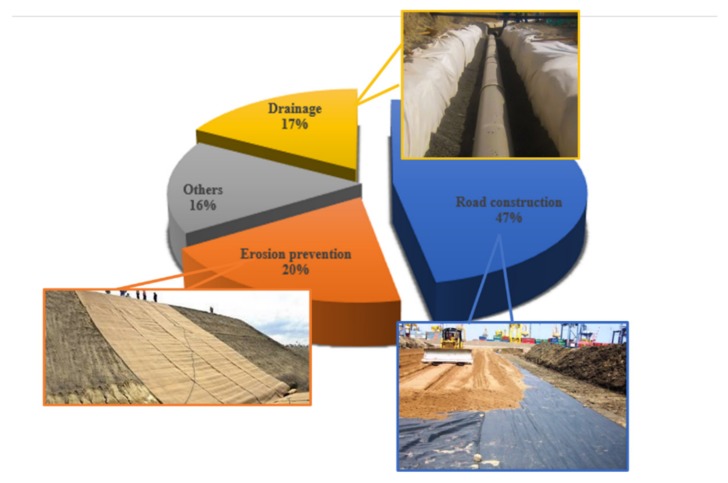
Global geotextile market share, by application, 2019 (%) [41,42,43].

**Figure 10 materials-13-01774-f010:**
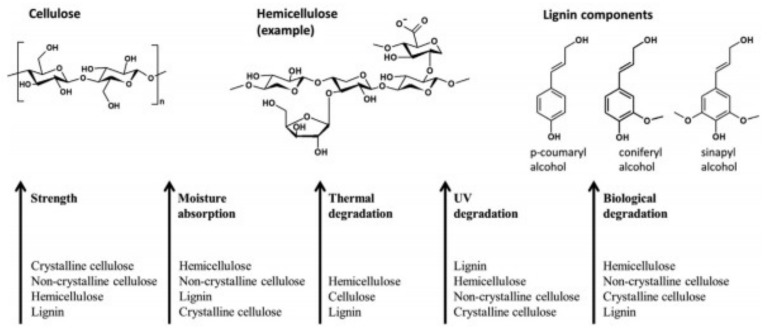
Chemical structure and properties of main components in plant fiber [5].

**Figure 11 materials-13-01774-f011:**
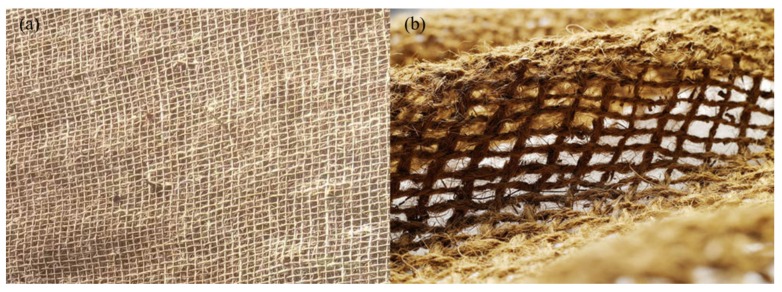
Common natural-fiber-based geotextiles: (**a**) jute geotextiles and (**b**) coir geotextiles.

**Figure 12 materials-13-01774-f012:**
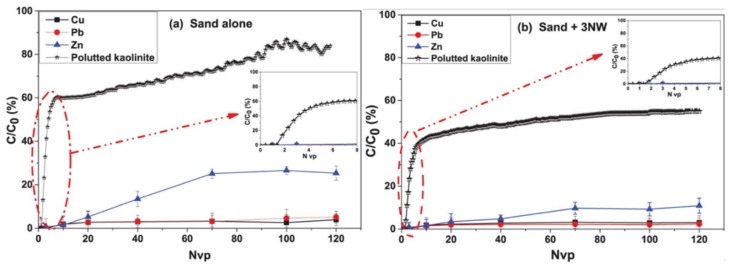
Soluble heavy metals breakthrough curves (BTCs): (**a**) sand alone and (**b**) sand + 3 flax geotextile [80].

**Figure 13 materials-13-01774-f013:**
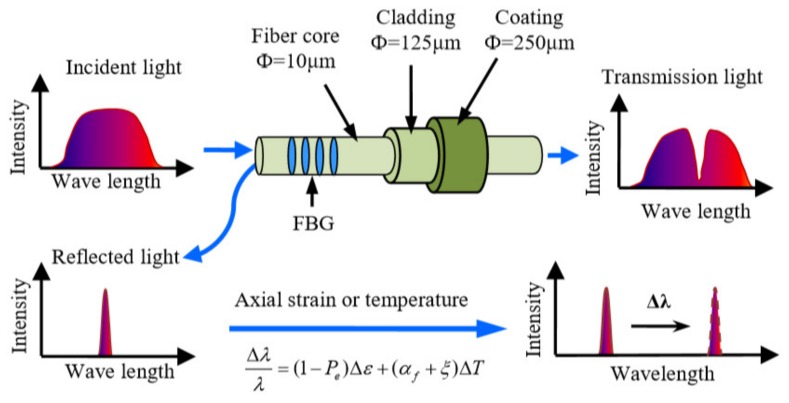
The working principle and sensing principle of Fiber Bragg grating (FBG) [104].

**Figure 14 materials-13-01774-f014:**
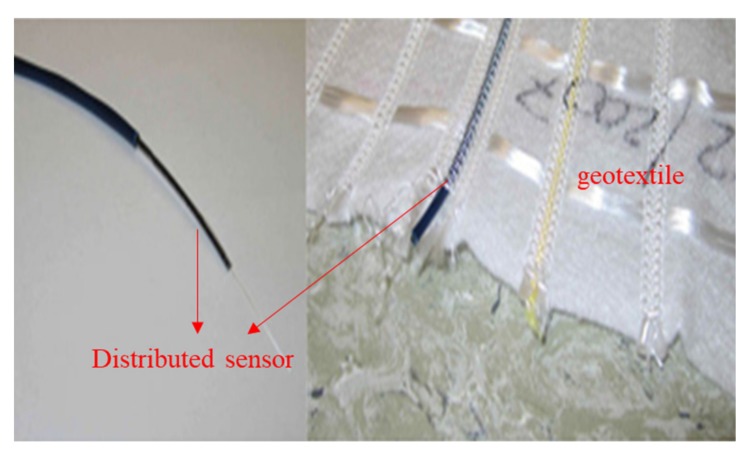
Intelligent geotextile based on distributed sensor.

**Figure 15 materials-13-01774-f015:**
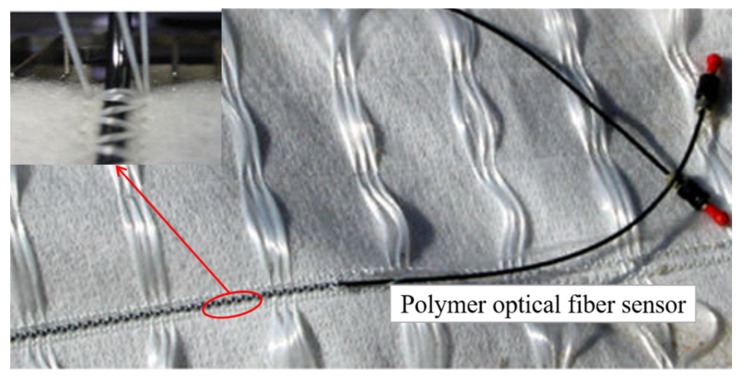
Intelligent geotextile based on polymer optical fiber sensor.

**Figure 16 materials-13-01774-f016:**
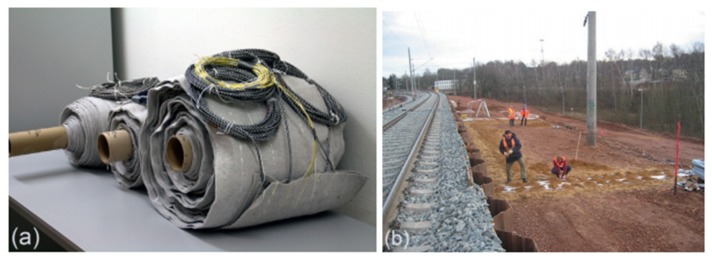
(**a**) intelligent geotextiles installed and (**b**) test site [113].

**Table 1 materials-13-01774-t001:** The properties of the most common polymers used as geotextiles [21,22].

Type of Fiber	Specific Gravity (kg/m^3^)	Modulus of Elasticity (GPa)	Tensile Strength (MPa)	Elongation at Break (%)	Acid/AlkAli Resistance	% of Total Synthetic Geotextile Production	Corresponding Geotextile Price ($/m^2^)
Polypropylene (PP)	910	1.5–12	240–900	15–80	High	~92	0.22–2.5
Polyethylene terephthalate (PET)	1400	2–2.5	45	120	Low	~5	0.15–2.0
Polyethylene (PE)	920–960	5–100	80–600	4–100	High	~2	0.11–1.2
Polyamide (PA)	1140	1–8.3	75–80	55–60	High	~1	0.27–1.5

**Table 2 materials-13-01774-t002:** Composition and properties of natural fibers commonly used to make natural geotextiles [55].

Type of Fiber	Cellulose (wt%)	Lignin (wt%)	Hemicellulose (wt%)	Density(g/m^3^)	Strain at Break (%)	Tensile Strength (MPa)	Young’s Modulus (MPa)
Flax	71–78	2.2	18.6–20.6	1.4–1.5	1.2–3.2	345–1500	27.6–80
Hemp	57–77	3.7–13	14–22.4	1.48	1.6	550–900	70
Jute	45–71.5	12–26	13.6–21	1.3–1.46	1.5–1.8	393–800	10–30
Kenaf	31–57	15–19	21.5–23	1.2	2.7–6.9	295–930	22–60
Ramie	68.6–76.2	0.6–0.7	5–16.7	1.5	2–3.8	220–938	44–128
Nettle	86	5.4	4	1.51	1.7	650	38
Sisal	47–78	7–11	10–24	1.33–1.5	2–14	400–700	9–38
Abaca	56–63	7–9	21.7	1.5	2.9	430–813	33.1–33.6
Cotton	85–90	0.7–1.6	5.7	1.21	3–10	287–597	5.5–12.6
Coir	36–43	41–45	0.15–0.25	1.2	15–30	175–220	4–6

**Table 3 materials-13-01774-t003:** Natural geotextiles used for protection and its protective effect.

Type of Geotextile	Application	Effect	Ref.
Palm-mat geotextiles	Prevent slope soil loss	Palm-mat geotextiles have obvious benefits of soil and water conservation	[57]
Palm-mat geotextiles	Prevention of soil erosion	Palm-mat geotextiles prevent erosion and repair gullies	[8]
Jute geotextiles	Prevention of river bed scour	It can effectively filter and separate, and prevent the migration of fine particles in free-flowing water	[58]
Coir geotextiles	Prevention of river bed scour	The effect of coir geotextiles as bank protection measure is excellent	[59]
Jute geotextiles	Prevention of river bed scour	The life of modified JGT is 600–700 days	[60]
Jute geotextiles	Prevention of river bed scour	The durability of jute geotextiles treated with environmental-friendly additives is up to 4 years, and the tensile strength meets the requirements of riparian application	[61]
Elephant grass, yorkgrass geotextiles	Prevention of soil erosion	Compared with the control, the soil loss of the two geotextiles decreased by 56.6% and 97.3% respectively, which was effective for erosion and sediment control.	[62]
water hyacinth, reed, sisal, Roselle geotextiles	Prevention of soil erosion	Sisal and Roselle have potential as raw materials of geotextile, while the fabric made of reed and water hyacinth is suitable for soil erosion control	[63]
Palm-mat geotextiles	Desert fixing sand	Palm-mat geotextiles have excellent water retention and erosion resistance	[64]
Coir, Jute geotextiles	Runoff control	The runoff control performance of jute geotextiles is better than that of coir geotextiles	[65]
Coir geotextiles	Prevention of soil erosion	After 12 months of exposure to environment, the tensile strength of untreated fibers remained at 23% of the initial strength	[65]
Jute geotextiles	Prevention of soil erosion	Jute geotextiles can delay runoff time, reduce runoff 62.1% and erosion 99.4% respectively.	[66]

**Table 4 materials-13-01774-t004:** Natural geotextile used for reinforcement and its effect.

Type of Geotextile	Application	Effect	Ref.
Coir geotextiles	Experimental study on bearing capacity	Coir geotextiles have great potential as reinforcement material	[67]
Kenaf geotextiles	Experimental study on bearing capacity	When kenaf geotextile are used in sandy soil, the bearing capacity of sandy soil is 414.9% higher than that of untreated soil	[68]
Jute geotextiles	Construction of rural roads	Compared with the part without geotextile, jute geotextiles can increase the road strength by 67–73%	[69]
Coir geotextiles	Reinforcement of Subgrade	By using coir geotextiles, CBR improvement factor can reach 1.5 to 2.2	[70]
Kenaf geotextiles	Reinforcement of full-size embankment on soft clay	Kenaf geotextiles can be used for short-term reinforcement of soil on soft clay	[71]
Jute geotextiles	Reinforcement of Subgrade	The long-term durability of jute geotextiles is not particularly crucial for the stability of subgrade reinforcement	[72]
Coir geotextiles	Reinforcement of soft soil foundation	Coir geotextiles have potential application value in strengthening soft soil foundation	[73]
Coir geotextiles	Reinforcement of Subgrade	The CBR value of subgrade increases by about 50%	[74]

**Table 5 materials-13-01774-t005:** Study on improving the properties of natural geotextiles.

Method	Type of Geotextile	Research	Effect	Ref.
Chemical modification	jute geotextiles	Esterification of Jute geotextiles	Stretching and chemical degradation resistance enhanced	[84,85]
	coir geotextiles	Durability studies of surface-modified coir geotextiles	The surface-modified geotextiles retained more than 70% of their initial tensile strength after burial in the top layer of soil after one year.	[87,88,89]
	jute geotextiles	Laccase treatment of Jute Geotextiles	Physical properties and surface hydrophobicity are improved	[90]
	jute geotextiles	Treatment of Jute Geotextiles with Isothiazolinone and Fluorocarbon Derivatives	Improvement of Antimicrobial and Water-proof Performance	[60]
	coir geotextiles	Lime-treatment	Lime treatment promotes the initial retention of cellulose in natural fibers	[11]
	Kenaf geotextiles	alkaline treated	Compared with untreated kenaf geotextile, the tensile strength of the geotextile treated with 6% NaOH is increased by 51.0%	[81]
Blending synthetic fiber	Nettle/PLA geotextiles	Tests on tensile strength, biodegradability and soil fertility enhancement	the geotextiles is promising for slope stabilization application	[82]
	jute/polypropylenegeotextiles	Treatment of Jute/PP Nonwoven Geotextile with Alkali	Tensile properties and puncture resistance were improved	[54,55]
	jute/polypropylenegeotextiles	Mechanical Properties and Damage Analysis of Jute/Polypropylene Nonwoven Geotextile	Compared with PP geotextiles, 40/60 jute/PP geotextiles have higher tensile strength and secant modulus	[91]

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
