# Peer review of "Review of Application and Innovation of Geotextiles in Geotechnical Engineering"

_materials, 2020, doi:10.3390/ma13071774_

Round 1

Reviewer 1 Report

The article present the review of application and innovation of geotextiles in geotechnical engineering. The topic of the article is in scope of journal. The manuscript contain appropriate description of present major trends in geotechnical engineering. However, in my opinion article must be slightly improved before publication. Some issues should contain more details. The following modification should be considered:

1. In the abstract part I suggest to add practical usage of geotextiles; please show the elements/object in civil engineering and reason, where this products could be effictively implemented.
2. Line 54: 'The selection of geotextile materials should be based on the actual situation of the project'  I recommend to choose 'should be based' to 'must be adequate to'. The geotechnical situation determine/force the
engineering solution.
3. Point 2.2. Please use capital letter in a word 'function'; the same in Table 4.
4. It will be beneficial to briefly describe phenomenon of suffosion in point 2.2.2 Geotextiles used in filtration.
Suffosion is a process involved with filtration.
5. Please add the reference to figure 9 if it is cited.
6. I suggest to remove word 'strain' in line 277; the strain are equal to mechanical deformation.
7. The text in figure 15 is illegible. Please improve it.
8. The case observed in the point 4.2.2. - it is worth to note more details about situation in Belchatow.
In case of Belchatow, very high values of displacement and velocity of landslide were observed.

Reviewer 2 Report

Congratulations to authors. Very interesting review regarding an important material used in infrastructures construction: geotextiles.

In general the paper is well presented but some information shall be clarified or completed:

  1. Figure 2: the geotextile used as separator shall have an high tensile strength, so maybe the Authors can emphasis this fact in this figure;
  2. Figure 3: in other situations a problem like the one shown in this image may have different causes. The Authors may include some other information in this figure or more clear images;
  3. Remove the hidden information included in some pictures, for example, in Figure 6;
  4. In lines 288/289 is written: “Secondly, its high cost and vulnerability also limit the use of this fiber-optic sensor in intelligent geotextiles [94,95]”. Can the Authors search and include any more information about costs and availability of this solution?
  5. Can the Authors better describe the examples reported in section 4.2?
  6. Before “6. Conclusion”, the authors may include another section with some information regarding one important aspect for the application industry: costs of each type of geotextile. Please, include at least a table with a percentage comparison between average direct costs of the most used geotextiles;
  7. Lines 405/406: “The application of nanofibers in geotextiles is a trend in the future”, that will bring other health problems for the productive industry.
